# A Zero-Shot Language Agent for Computer Control with Structured Reflection

**Tao Li**[⋆]  **Gang Li**[⋆]  **Zhiwei Deng**[⋆]  **Bryan Wang**[◇†]  **Yang Li**[⋆]

[⋆]Google Research, Mountain View, U.S.A.

[◇]University of Toronto, Ontario, Canada

{tlinlp,leebird,zhiweideng,liyang}@google.com

bryanw@dgp.toronto.edu

## Abstract

Large language models (LLMs) have shown increasing capacity at planning and executing a high-level goal in a live computer environment (e.g. MINIWOB++). To perform a task, recent works often require a model to learn from trace examples of the task via either supervised learning or few/many-shot prompting. Without these trace examples, it remains a challenge how an agent can autonomously learn and improve its control on a computer, which limits the ability of an agent to perform a new task. We approach this problem with a *zero-shot* agent that requires no given expert traces. Our agent plans for executable actions on a partially observed environment, and iteratively progresses a task by identifying and learning from its mistakes via self-reflection and structured thought management. On the easy tasks of MINIWOB++, we show that our zero-shot agent often outperforms recent SoTAs, with more efficient planning. For tasks with more complexity, our reflective agent performs on par with prior best models, even though previous works had the advantages of accessing expert traces or additional screen information.

## 1 Introduction

Prior works have shown promises in using large language models (LLMs) for action generation, e.g. SAYCAN (Brohan et al., 2023), REACT (Yao et al., 2023), TOOLFORMER (Schick et al., 2023), and SWIFTSAGE (Lin et al., 2023)) over a variety of live environments (e.g. MINIWOB++ (Shi et al., 2017; Liu et al., 2018), ALFWORLD (Shridhar et al., 2021), and ALPHACODE (Li et al., 2022). A shared approach is to use LLMs to follow expert traces, comprehend environment changes, plan future actions, and execute an action by composing API calls; all in the form of text. Some works have shown that iteratively attempting a task with

several rounds of self-reflection can substantially improve task completion, e.g., REFLEXION (Shinn et al., 2023), SELF-REFINE (Madaan et al., 2023). During the process, LLMs are prompted to update a prior execution plan according to feedback from the environment. Such updates become part of the prompt for the action generator in the next round.

Recently, MINIWOB++ has been used as a testbed for LLM's capacity at modularized computer tasks. To learn a task, a common approach is the use of extensive trace examples of the task for direct supervision (e.g., CC-Net (Humphreys et al., 2022), WebGUM (Furuta et al., 2023)), self-supervision (Gur et al., 2023), or few/many-shot prompting (e.g., RCI (Kim et al., 2023), SYNAPSE (Zheng et al., 2023)). They have achieved more than 90% task completion rate on dozens of computer tasks, seemingly to have solved the computer control problem.

However, the requirement of expert traces for learning to perform a task limits the agent's ability on new tasks. Without using carefully selected traces as guidance, *can an agent autonomously learn and improve its control on a computer?* To address this question, we propose a zero-shot agent. We build our agent on top of PaLM2 (Anil et al., 2023), a recent LLM, and our agent employs a unified instruction prompt set across different tasks, without extensive tailoring for individual tasks.

In addition, recent works, e.g., RCI (Kim et al., 2023), ADAPLANNER (Sun et al., 2023), and SYNAPSE (Zheng et al., 2023), utilize a screen representation that may include much more information than what is presented to a user on the screen. For instance, Fig. 1 shows an example of elements that are not shown on the screen yet present in the HTML that is fed to the LLM. Using such additional information arbitrarily reduces the difficulty for the agent to perform the task. Yet in general use cases, such information might not be readily available, and relying on such information can po-

---

[†]Work done during internship at Google Research.

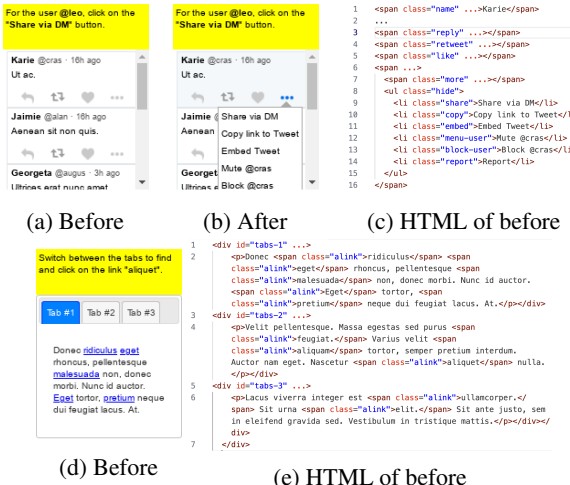

|         |        |         |         |
|---------|--------|---------|---------|
| (a) Before | (b) After | (c) HTML of before | |
| (d) Before | | (e) HTML of before | |

Figure 1: Inconsistent screen representations. Fig. 1a-1c: before and after clicking the "more" button on the *social-media* task (seed=2). HTML before the clicking already revealed the content. Fig. 1d-1e: Similar issue in the *click-tab-2* (seed=0).

tentially hamper the applicability of the agent. We manually examined 13 relatively challenging tasks on MINIWOB++ that are supposed to span multiple screens, and found 5 of them contained such information—multi-screen information in a single observation—in their HTML.

Our contributions are as follows: Firstly, we employ a compact screen representation that assumes much less information than what is used by previous works, thus resulting in a more general and realistic test environment. Secondly, we propose a simple yet efficient action planner that can accurately plan out executable actions on a state in one pass. With recent capacity of LLM, we show that such a "naive" strategy can solve almost all the easy tasks on the MINIWOB++ benchmark. For more challenging tasks, we take inspiration from Reflexion (Shinn et al., 2023) and propose a structured thought management strategy to facilitate reflection, allowing the agent to effectively learn and improve from exploration failures. With a few rounds of attempts, our agent achieves comparable performance with prior few/many-shot state-of-the-art. To the best of our knowledge, our agent is *the first zero-shot design for computer control tasks*[1].

## 2 Background

LLMs have become an emerging tool for planning and executing necessary steps to fulfill a top-level goal. These models have exhibit high capacity to follow in-context traces to solve complex tasks.

**Planning & Reflection** REACT (Yao et al., 2023) used intermediate thoughts to guide long-chain of action planning in a live environment. Beyond one trial of planning, REFLEXION (Shinn et al., 2023) and SELF-REFINE (Madaan et al., 2023) recently found the ability of LLM to self-criticize and self-improve can iteratively learn and solve a given task via multiple trials. Nevertheless, recent planning works require extensive and customized trace prompts for the LLM agent to learn accurate planning. SWIFTSAGE (Lin et al., 2023) reduces the extensive planning calls to LLM by training a small and fast planning module to facilite long-chain of actions. In a similar motivation, our zero-shot agent is based on an efficient staged planner. We list a detailed comparison in Tab. 1.

| Planning | Unsup | Trace | Efficient | Feedback | Reflection mem |
|----------|-------|-------|-----------|----------|----------------|
| RCI | ✓ | ✓ | ✗ | sparse | 1 |
| ADAPLANNER | ✓ | few | ✓ | detailed | — |
| REFLEXION | ✓ | few | ✗ | detailed | 3 |
| SWIFTSAGE | ✗ | — | ✓ | detailed | — |
| Ours | ✓ | 0 | ✓ | sparse | N |

Table 1: Comparison with prior work on reflective planning. N: number of actions to complete the task. Efficient ✗: requires planner call for each action.

| MINIWOB++ | Unsup | Trace | Efficient | Consistent Screen |
|-----------|-------|-------|-----------|-------------------|
| CC-NET | ✗ | large | ✗ | —[2] |
| PIX2ACT | ✗ | large | ✗ | ✓ |
| RCI | ✓ | few/many | ✗ | ✗ |
| ADAPLANNER | ✓ | few | ✓ | ✗ |
| Ours | ✓ | 0 | ✓ | ✓ |

Table 2: Comparison with prior work on MiniWoB. For discussion on consistent screen, please refer to Sec. 3.1.

**Language/Vision Agent for Computer Control** MINIWOB++ (Shi et al., 2017) has several dozens of fundamental computer tasks hosted as live environments. Recent advances on this benchmark have benefited from extensive human annotation to facilitate behavior cloning and reinforcement learning, such as CC-NET (Humphreys et al., 2022) and PIX2ACT (Shaw et al., 2023). Beside these models that rely on multimodal or vision input, another

---

[1]Code and notebook: https://github.com/google-research/google-research/tree/master/zero_shot_structured_reflection

[2]No public source for CC-NET's screen representation at the time of writing this paper.

track is using LLMs as an off-the-shelf agent, and use prompted inference for action generation, such as RCI (Kim et al., 2023), ADAPLANNER (Sun et al., 2023), and SYNAPSE (Zheng et al., 2023). We highlight our technical differences in Tab. 2.

# 3 Environment Interface

The role of a language agent is to comprehend screen representations (Sec. 3.1&3.2), execute actions on the environment (Sec. 3.3), and react to environment feedback (Sec. 3.4).

## 3.1 Treatment of Screens

The definition of a screen observation varies by modality in recent works. The screen observation for a vision-based model (e.g. Humphreys et al., 2022; Shaw et al., 2023) can be constrained by various viewport configurations. For a language model, specifically those taking in HTML code, a screen observation is often the page source of a screen, e.g., ones instantiated from a MINIWOB++ task template with a given random seed. When HTML elements are not constrained by viewport configuration, the need for scrolling action is gone. However, as discussed in Sec. 1, we do not immediately use the expanded HTML if the expansion requires a UI action: we only expand the HTML representation when the agent actually takes the action. The design relaxes the assumption about the environment, and forces the agent to learn to behave rationally when it is given limited information.

## 3.2 Compact Screen Representation

Raw HTML code tends to be verbose, which poses a practical challenge for LLMs that often have an inherent limit on the input or context length. Zheng et al. (2023) designed a technique for structured prompting example selection to extract more informative trace examples, as a way to reduce the input content to LLMs. MindAct (Deng et al., 2023) ranked and consolidated element of interests to represent web page snippet. Alternatively, we take inspiration from Wang et al. (2023a) to heuristically simplify the HTML code of each screen, retaining key attributes for each leaf element, i.e., id, class, text, placeholder, value, and position on a 3x3 screen grid. Such simplification has shown to give compelling results on UI understanding tasks. An example is shown in Fig. 2.

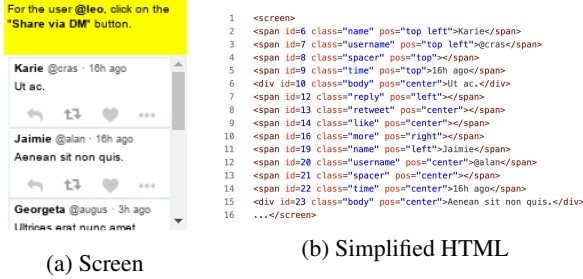

(a) Screen      (b) Simplified HTML

Figure 2: Example of compact screen representation.

## 3.3 Action Space

For each action, our agent model outputs commands in a format that is specific to the action type. Specifically, we use three types of actions as shown in Tab. 3. Recent LLMs such as PaLM-2 (Anil et al., 2023) are good at following such an output format. More prompt details are given in Appx. A. To deterministically ground an action command on MINIWOB++ environment, we follow the approach in prior language-only works (e.g. Kim et al., 2023) to access HTML elements by XPATH pattern. When grounding click actions on the actual environment, we use the compact element id (Sec. 3.2) which is aligned to the actual HTML element in the raw HTML. For the type action, we decompose it into a click action followed by a series of keyboard presses for the text.

| click | type | special key |
|---|---|---|
| click id=6 | enter "text" to id=10 | press ARROWDOWN x N |

Table 3: Action types and example commands.

## 3.4 Environment Feedback

MINIWOB++ differs from TEXTWORLD-like environment (Shridhar et al., 2021) in that state change from taking an action is not naturally phrased out. Instead, an agent will need to observe the entire screen change implicitly, making it less convenient for the agent to adopt Chain-of-Thought (Wei et al., 2022) reasoning. We broadly categorize trial ending conditions into: 1) correct, 2) cycle, 3) no change, 4) incomplete, 5) exception, and 6) failed. Condition 2) and 3) compare the HTML code of the current screen to those of prior screens. Condition 5) happens when a grounding action can not be successfully executed. During multiple trials, each ending condition is associated with a prompt for the reflection inference.

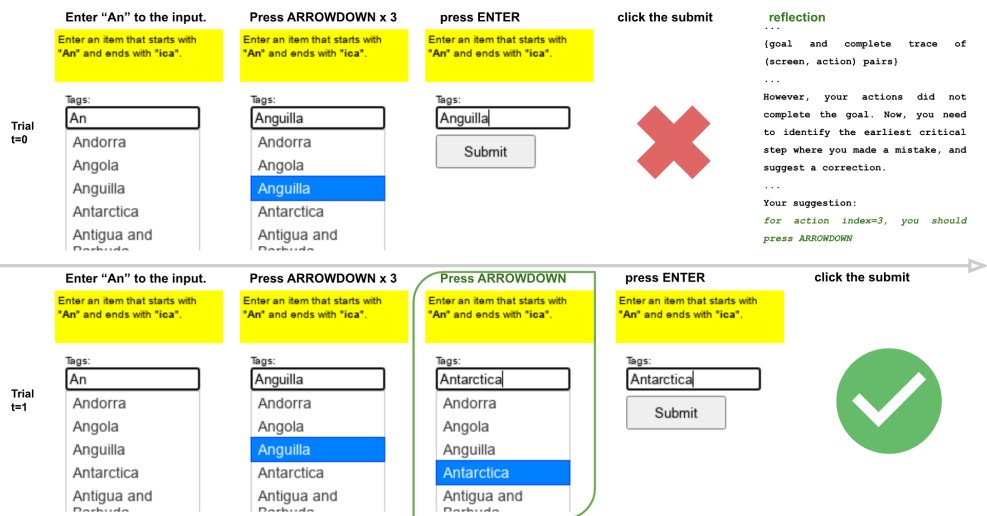

Figure 3: An example of successful reflection trial by our zero-shot agent on MiniWoB++ task *use-autocomplete* with seed=0. Step actions are paraphrased from the actual executable ones for readability.

## 4 Planning Strategies

In this section, we summarize the planning strategies used in recent LLM-based planning works to motivate our staged planning. With a given goal, an agent model is to issue actions based on prior interaction with the environment. For brevity, let us denote the interaction as a sequence of state and action pairs $(s_i, a_i)$.

### 4.1 Iterative Planning

In iterative planning (e.g. Yao et al., 2023; Madaan et al., 2023), the agent model loops over generating an "atomic" action $a_i$, grounding it on the environment for execution, and then observing for the next state $s_i$. That is,

$$a_i \sim \tau_\theta(a|s_i, a_{i-1}, s_{i-1}, ...a_0, s_0) \qquad (1)$$

where $\tau_\theta$ denotes the planning model. Such planning is a common choice for environments that require observation by exploration. With responsive environments (e.g. Côté et al., 2018; Shridhar et al., 2021), such an agent can benefit from a long history of interaction that can be easily connected to Chain-of-Thought reasoning (Wei et al., 2022).

### 4.2 Plan-Then-Adapt

Recently, Kim et al. (2023) observed that an initial, and yet rough, plan could help iterative planning. Formally,

$$(a_0, a_1, ...a_n) \sim \tau_\theta(a|s_0) \qquad (2)$$
$$\bar{a}_i \sim z_\theta(\bar{a}|s_i, \bar{a}_{i-1}, s_{i-1}, ..., (a_0, a_1, ...a_n)) \qquad (3)$$

where $z_\theta$ adapts those initial steps to be executable actions ($\bar{a}$'s) on the environment. In practice, both $\tau_\theta$ and $z_\theta$ use the same LLM.

Conceptually, this is similar to zero-shot planner (Huang et al., 2022) and ReAct (Yao et al., 2023) that intermediate thought can help plan long chain of actions. The downside though is that the agent needs to follow carefully crafted few-shot trace examples to make a good initial plan. AdaPlanner (Sun et al., 2023) addresses this issue with an adaptive plan refiner that monitors the state-action compatibility and issues refined actions when there is a mismatch. This line of planning often needs to deal with hallucinations in the initial plan since, after all, the agent model only observes $s_0$ but needs to plan for unobserved states.

### 4.3 Staged Plan-And-Follow

Prior works essentially add on extra planning components to the agent. Instead, we adopt a simpler planning strategy. For computer environment, agents often sees a state where multiple actions can be executed on, without the need to observe nuanced state changes, e.g., multiple selection on a list. In such cases, iterative planning on a single screen can be less efficient, and often, unnecessary. On the other hand, *plan-then-adapt* generates actions beyond executable ones that could confuse the LLM agent during the adaptation step. Furthermore, both approaches require the agent to iteratively generate the next action, requiring an LLM to have a large context window.

To address these issues, we take a step in the mid-

dle by maximally planning actions that are visible on the current state all at once. After the planning, the agent only needs strictly follow the generated plan, and such process repeats over multiple screens. Formally,

$$(a_i^0, ...a_i^k) \sim \tau_\theta(a|s_i, \mathbf{a}_{i-1}, \mathbf{a}_{i-2}, ...\mathbf{a}_0) \quad (4)$$

where each stage is essentially generating $k$ executable actions for state $s_i$. Note that, we also omit former states in Eq. 4 to make inference more efficient. In practice, we found that a simple statement, in natural language, that summarizes what functionality is achieved by the action, is a good representation for the state-action pair $(s_i, a_i)$.

**Implementation details.** We once again rely on the underlying LLM to follow execution instructions as in Eq. 4. Prompt details are in Appx. B-E. In rare occasions, agent model predicts fewer steps (e.g., forgetting to submit) or more steps (e.g., hallucinating non-executable actions) than needed. For the former, we loop over the planning in Eq. 4 until no further plan is generated. For the latter, we halt the execution of the current plan and resort to self-reflection for correction.

## 5 Structured Self-Reflection

In practice, a high-level human instruction or goal can be ambiguous, and an environment can be partially hidden. Therefore, agents are prone to making mistakes; this is even true for a human user when executing a task (Humphreys et al., 2022). Once a negative signal is given, such as *no change* or *failed* (Sec. 3.4), we ask the agent to reflect on its past action trajectory, suggest an improved version, and then retry the task. An example of successful reflection from our agent is shown in Fig. 3.

In recent works (Shinn et al., 2023; Madaan et al., 2023), reflection is conducted at the end of each trial by accumulating a text entry. The entry is essentially a natural language statement about what should have been done instead. At trial $t$,

$$a_i \sim \tau_\theta(a|s_i, a_{i-1}, ...a_0, s_0; R_t) \quad (5)$$

$$R_{t+1} \sim \text{REFL}_\theta(a_n, s_n, ...a_i, s_i, ...; R_t) \quad (6)$$

where $R_t$ consists of a list of $(a_i, a_i')$ pairs, each denotes to update the wrong action $a_i$ to $a_i'$. In the followup trial $t + 1$, accumulated entries in $R_{t+1}$ are prefixed to the prompt for the agent planner $\tau_\theta$.

The amount of entries maintained in the reflection memory is limited by multiple factors. For one,

it increases the input length of LLM. Moreover, it requires the LLM agent to handle the thought structures through multiple trials. In practice, Shinn et al. (2023) limited the memory size $\in [1, 3]$.

### 5.1 Structured Thought Management

In a zero-shot setting, reflection puts a heavy burden on LLM's capacity to follow a complex instruction set (in addition to those in Sec. 3.3&4.3). After all, in this case, LLM has no expert traces to follow, thus need to learn from trials. With increasing the number of trials, reflection memory essentially forms a combinatorial problem for the agent to solve. For a time step $i$, there can be multiple failed actions in historical trials, thus should be avoided. For a trial $t$, if the agent identified a critical mistake at time $i$, reflections on later time steps can be considered outdated.

In our preliminary study, we found it is important to devise a way to help an LLM agent maintain this memory. Otherwise, when a reflection entry $(a_i, a_i')$ is given, even a state-of-the-art LLM can still 1) repeat the same mistake $a_i$, 2) fail to follow the reflected plan to even reach $s_i$, and 3) bounce between wrong actions it collected in prior trials.

To make reflection run more reliably and efficiently, we propose a structured self-reflection in Algo. 1. When a suggested action $a_i'$ is given by the reflection agent, we enforce our agent to plan exactly $a_i'$ at step $i$. Moreover, to avoid looping over two failed actions at the same step, we use a disabled action set $D$ to memorize them and jointly disable these actions in the environment. Finally, we clear reflection entries for future steps if an early entry is updated. With this management, our agent is no longer bounded by LLM's input limit, and has a memory size $N$.

---

**Algorithm 1:** Structured Thought Management
| |
|---|
| 1: $R = [\varnothing] * N; D = [\varnothing] * N;$ |
| 2: for $t \in [0, T)$: |
| 3:    for $i \in [0, N)$: |
| 4:      if $R[i]$ and $R[i].a' \notin D[i]$:   // if has reflection |
| 5:        $a_i = R[i].a'$          // action from reflection |
| 6:      else: $a_i \sim \tau_\theta(a|...)$     // regular planning |
| 7:    if *needToReflect*:        // if error happens |
| 8:      $(a_j, a_j') \sim \text{REFL}_\theta(...)$    // reflect |
| 9:      if $R[j] \neq \varnothing$: |
| 10:        $D[j]$.add($R[j].a$)     // record wrong click |
| 11:      $R[j] = (a_j, a_j')$       // record verbal reflection |
| 12:      $R[j + 1 :] = \varnothing; D[j + 1 :] = \varnothing$   // clear mem |

---

Note that in line 6, we use the staged planner in Eq. 4.3 which does not depend on the iteratively up-

dated $R$, thus is different from recent works (Eq. 6).

**Interplay with Staged Planning.** Suppose the staged planner predicted $[a_0, ...a_i, ...a_n]$ but execution had failed, and the reflection step identified $a_i$ as the earliest mistake, thus suggested $a_i'$. In the next trial, we will repeat the executions from $a_0$ to $a_{i-1}$[3], and intercept the agent planner at step $i$ to enforce the execution of $a_i'$. For steps after $i$, we bring our planner to the next stage. In the worst case where an agent fails at every step, our staged planning essentially falls back to the *plan-then-adapt* (Sec. 4.2), except having no initial plan.

## 5.2 Constraining Action Space

For an updated action $a'$ at reflection trial $t$, we enforce it to be taken at the associated time step if and only if $a'$ is not an failed attempt before trial $t$. It can be tricky to prompt LLM to follow such simple combinatorial constraint in text, especially as a mixture of positive and negative signal surrounded by other instructions. Therefore, we found it is crucial to explicitly disable those previously failed actions in the corresponding screen representation. This, however, does not mean removing the corresponding element from the HTML pseudo code. We instead only remove the id attribute, and still allow the element information to be presented to LLM. We only do so for click-type actions.

For non-click actions, the disable set $D$ cannot be easily enforced on the environment and the LLM agent. We can indeed prompt the underlying LLM saying certain special keys are invalid or certain texts not to input. However, we did not observe a positive impact from doing so in our preliminary experiment[4]. Thus, we fallback to only deterministically generate the $a_i'$ at time step $i$.[5] We locate the time step by prompting the reflection agent to output in format: "For action index=$i$, you should $a_i'$". This differs from prior work (Shinn et al., 2023) which uses reflection memory as sticky entries in LLM prompts across all time steps.

## 6 Experiments

We start with categorizing tasks by their planning complexities to have an isolated testbed. Then we

---

[3]Up to this point, the work flow is similar to the refine-then-resume in ADAPLANNER (Sun et al., 2023).

[4]A possible reason is that the instruction set in LLM prompt is already dense and reflection prompt tends to be long, thus such nuanced requirements sometimes get ignored.

[5]The downside is that agent can potentially loop over two non-click actions across multiple reflection trials.

experiment with our staged planning in Sec. 6.3-6.4. Finally, we examine if our zero-shot agent can learn from mistakes in Sec. 6.5. Our prompts are in Appx. A-E. Complete results are in Appx. F.

## 6.1 Setup

We focus on 43 MINIWOB++ tasks that are suitable for evaluating language-based models. This differs from prior work since we excluded those 1) require visual signal to solve (e.g., *count-shape* and *grid-coordinate*); and 2) expose insufficient language API to operate (e.g., *enter-date* and *enter-time*); The motivation for this filtering is simple: even if some filtered tasks can be solved by an LLM agent, it does not generalize. Furthermore, we do not include *terminal* as the synthetic console supports a very limited set of commands while the LLM, in our preliminary experiment, tends to use more sophisticated ones.

We separate these 43 tasks into three categories: 1) 1-screen-1-step, 2) 1-screen-$n$-step, and 3) $n$-screen-$n$-step. If the task involves state update (e.g. expanding dropdown list or openning hidden tab), the task is *n-screen*. If the task can be solved by just one action, it is *1-step*; otherwise *n-step*. The task distribution is reported in Tab. 4.[6]

For each task, we evaluate with 25 different random seeds, starting from seed=1000, similar to Pix2Act (Shaw et al., 2023). Performances are reported as the correct completion rate over multiple runs. For validation and prompt design, we use seed $\in [0, 10]$. For the LLM agent, we use the FLAN-PaLM2 L (Anil et al., 2023) with temperature 0 across all evaluations for better reproducibility.

|  | 1-screen-1-step | 1-screen-n-step | n-screen-n-step |
|---|---|---|---|
| #Task | 10 | 20 | 13 |

Table 4: Task distribution for each category in MINIWOB++.

## 6.2 Model Comparison

For each task category, we compare with prior best models that rely on language as input signal, including supervised models, i.e., WEBN-T5 (Gur et al., 2022) and CC-NET (Humphreys et al., 2022), and agents based on prompted inference, i.e., RCI (Kim et al., 2023) with GPT-3.5

---

[6]Based on our categorization, the screen issue (Sec. 1) impacts the *n-screen-n-step* category.

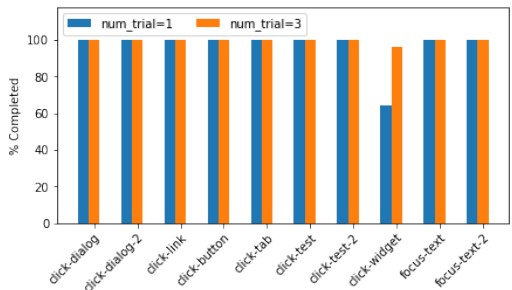

Figure 4: Performance on 1-screen-1-step tasks. *click-widget* contains ambiguous task objective, thus reflection helps.

## 6.3 Single Step Tasks

We compare our zero-shot agent on the easiest category (1-screen-1-step) tasks against recent state-of-the-art. As shown in Fig. 4, our agent achieves 100% accuracy on correctly finishing 9 tasks, even without reflection. One exception is the ambiguous *click-widget* which, without in-context trace prompt, can be easily failed. For instance, the task could ask agent to click on text widget, however, input text and text area are not deemed correct. With 3 rounds of reflection trials, our agent achieved 96% completion rate on it. Overall, we have 96.4% with 1 trial, and 99.6% with 3 trials. In comparison, with few-shot trace examples, RCI (Kim et al., 2023) achieved 99.8% with 1 round of reflection (at the plan level).

## 6.4 Iterative Planning v.s. Staged Planning

We compare these two approaches using 1-screen-$n$-step tasks. We hope these experiments can answer that, with a given state, whether one should query the agent for actions one at a time or once for all. We compare the prior state-of-the-art works with our staged planning in Tab. 5, showing that one can simply plan out all executable actions on a screen and "blindly" execute them. Doing so can substantially reduce LLM queries and still achieve high completion rate.

We report detailed completion rate on all 20 *1-*

| Supervised | | Few/N-shot | | Zero-shot (Ours) | |
|---|---|---|---|---|---|
| WebN-T5 | CC-Net | RCI | AdaPln | $T = 1$ | $T = 3$ |
| 60.4 | 95.1 | 96.1 | 96.5 | 95.3 | **97.3** |

Table 5: Average performance on *1-screen-n-step* tasks, 16 shared across all models. T: number of trials.

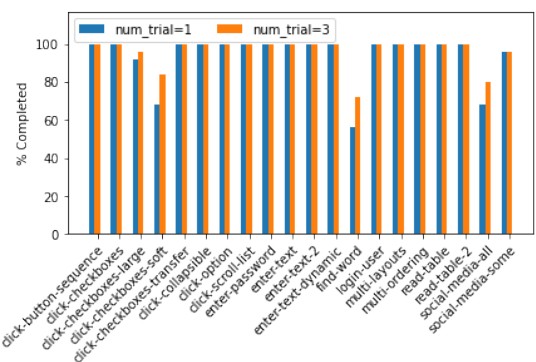

Figure 5: Performance on 1-screen-n-step tasks.

*screen-n-step* tasks in Fig. 5. Our agent achieved 94.0% completion in 1 trial, and 96.2% in 3 trials.

## 6.5 Reflective Planning on Challenging Tasks

Here, we move on to more challenging (*n-screen-n-step*) tasks to show the impact of our efficient reflection. Task-wise completion rate is reported in Fig. 6. Firstly, we observe without examplar traces, zero-shot agent tends to fail on the first trial. This happens often in tasks that requires exploring across multiple screens, e.g., *click-menu-2*, *click-tab-2-hard*, and *search-engine*. After a few rounds of exploration, our agent achieved substantially better completion rate by avoiding previous negative signals recorded in the memory. Our agent continues to improve even with $T = 5$, suggesting more efficient reflection than prior work e.g., RCI only capable of one round of reflection at plan level.

| Supervised | | Few/N-shot | | Zero-shot (Ours) | |
|---|---|---|---|---|---|
| WebN-T5 | CC-Net | RCI | AdaPln | $T = 1$ | $T = 5$ |
| 31.0 | **97.2** | 85.8 | 89.7 | 73.5 | 87.3 |

Table 6: Comparison on 11 shared tasks across different models in the *n-screen-n-step* category. T: number of trials.

Again, we compare with prior best models in Tab. 6. The few-shot models exploited inconsistent screens (as discussed in Sec. 1), thus our work is in an unfair disadvantage against them. Despite such disadvantage, our agent achieved performance comparable to them. Importantly, our agent does not re-

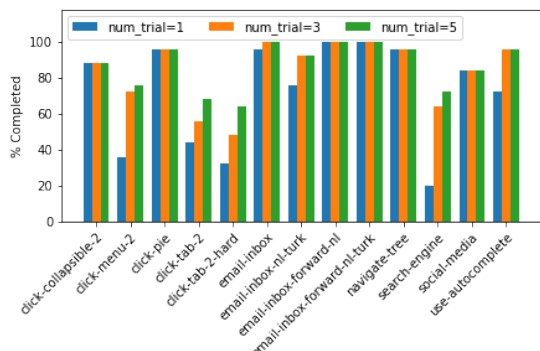

Figure 6: Performance on n-screen-n-step tasks.

quire in-context trace examples for few-shot, sometimes many-shot, prompting, and no customized and detailed environment feedback. Finally, we note that the gap on complex tasks between supervised model and unsupervised ones is still large.

## 7 Analysis & Discussions

### 7.1 Ablation on Reflection Strategies

Here, we compare our structured reflection against the "original" reflection mechanism. We should note that reflection is a general scope that has different formations (e.g. Shinn et al., 2023; Madaan et al., 2023) and was introduced on environments (e.g., ALFWORLD) that are significantly different from MINIWOB++. Moreover, it was often used along with iterative planning strategy, which is not directly compatible with our staged planning.

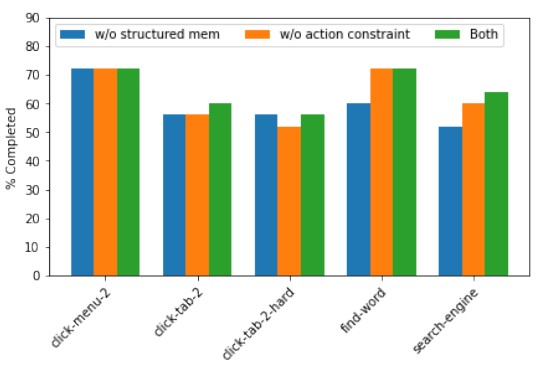

Figure 7: Comparison of reflection strategies with $T = 3$.

Therefore, we use an adapted version for comparison: an agent that uses structurally managed timestep[7] while structurally thought management[8]

---

[7]We insert reflection thought at the corresponding time step so that actions before this time step can be deterministically replayed for better efficiency.

[8]On top of the structurally managed timestep, we also manage the expiration of thoughts over multiple trials, as well

is turned off. This setting is the comparison between *Both* v.s. *w/o structured mem* in Fig. 7 where we select 5 challenging tasks and run 25 seeds for each setting. Clearly, our structured reflection is a beneficial add-on.

### 7.2 Ablation on Action Constraint

A useful technique we proposed in Sec. 5.2 is to delete the id field in the HTML pseudo code to heuristically discourage LLM agent from issuing the corresponding action. This is essentially minimal change to the input. In Fig. 7, we ablate on this small change by comparing *Both* v.s. *w/o action constraint* and show that it is beneficial to apply the action constraint.

### 7.3 Statistical Significance over Trials

We evaluate statistical significance across various trials on the *n-screen-n-step* tasks. For each task, we consider all 25 example predictions. This gives us $13 \times 25 = 325$ samples for each comparison. Using t-test (Dror et al., 2018), the results are indeed significant ($p < 0.05$) as shown in Tab. 7. For task-wise significance, see Appx. G.

| Baseline | Hypothesis | p-value |
|---|---|---|
| T=1 | T=3 | 2e-10 |
| T=3 | T=5 | 0.002 |
| T=1 | T=5 | 2e-12 |

Table 7: Significance test using t-test comparing different number of trials.

### 7.4 Planning Call Reduction

In Tab. 8, we highlight the efficiency boost by suing our staged planning formulation. We illustrate the result on *1-screen-n-step* tasks that require relatively long action traces ($\geqslant 7$ actions) on a single screen, and compare the number of planning calls for completed traces as well as failed traces.

### 7.5 Compact Screen & Input Length Limit

Representing user interface as HTML puts a high demand on LLM's context capacity. One instance is the *social-media-all* task that can span more than a dozen candidates, each with multiple options. As a result, flattening the complete set of state-action pairs can easily run over the input limit for the reflection agent, since it needs to observe the entire trace. On this task, we noticed that nuanced

---

as constraining action space.

| | T=1 (Success-only) | | | T=3 | | |
|---|---|---|---|---|---|---|
| Task | IP | SP | ↓ | IP | SP | ↓ |
| click-checkboxes-large | 234 | 24 | 89.7% | 270 | 31 | 88.5% |
| click-checkboxes-soft | 81 | 19 | 76.5% | 167 | 64 | 61.7% |
| multi-layout | 175 | 25 | 85.7% | 175 | 25 | 85.7% |
| click-checkboxes-soft | 114 | 20 | 82.5% | 224 | 64 | 71.4% |

Table 8: Planning call reduction by staged planning. Comparisons are on the successful first trials and all trials when $T = 3$, using 25 examples per task. **IP**: number of planning calls required for iterative planning. **SP**: number of planning calls in staged planning. ↓: percentage of planning calls reduced by staged planning.

actions does not substantially change the screen. Therefore, we always stick to the first screen when constructing prompt for the reflection agent. A more autonomous method can be *state filtering* in SYNAPSE (Zheng et al., 2023).

Lining up the separation of what HTML elements to expose to LLM is important for evaluation. As we have seen that many of the MiniWoB++ tasks are already easy for today's LLM. Exposing more unseen elements risks hiding the actual challenge in the navigation tasks. For instance, exposing unseen elements basically simplifies n-screen-n-step tasks into 1-screen-n-step ones. However, our experiment shows that n-screen-n-step ones are actually much harder to deal with.

### 7.6 Capacity of Staged Planning

To better understand the planning capacity and limit of our staged planning, we experiment with *1-screen-n-step* tasks that have extensive number of candidates. Namely, we use *click-checkboxes* and *social-media* as probe tasks, and report in Tab. 9. Both tasks are multi-choice, but differ in their candidate structure complexity.

| Task | #Gold | Completion rate |
|---|---|---|
| click-checkboxes | < 10 | 100 |
| click-checkboxes | ≥ 10 | 90 |

| Task | #Candidate | Completion rate |
|---|---|---|
| social-media-all | < 10 | 80 |
| social-media-all | ≥ 10 | 40 |

Table 9: Impact of number of candidate/gold actions on task completion. We evaluated 20 examples for each setting, and $T = 1$.

For the *click-checkboxes* task, we separate examples by their number of actions required[9]. The

---

[9] which can be heuristically parsed from the task command.

screen representation for this task is relatively simple as each checkbox corresponds to a line of text. This differs from the *social-media* task where each candidate has multiple actionable, sometimes ambiguous, items, thus putting a stronger requirement to LLM for disambiguation. We observe a pattern in Tab. 9 that with flat and less ambiguous screen, LLM has high capacity to accurately plan out multiple steps in one inference call. In such case, one could just execute all planned steps without needing repetitive planning calls. But with complex screen constructions, the capacity of one-pass planning is reduced by a large margin. Prior work (i.e. RCI) constrained the number of candidates in the *social-media* task to $[3, 6]$. We observe that relaxing such constraint introduces significant difficulty for planning. Therefore, multiple trials of reflection can help the agent in these complex scenarios.

### 8 Conclusions

We proposed the first zero-shot agent for computer control tasks. Our agent design generalizes the workflow for easy and complex tasks via efficient planning and structured thought management. We evaluated our agent on the MINIWOB++ benchmark, showing that our agent, with often one pass of planning query, outperforms the best iterative planning agent as well as supervised state-of-the-art on simple tasks. For complex tasks, we show that our agent design performs on par with the best LLM-based model via more efficient planning and reflection, without requiring manually crafted trace prompts and ad-hoc environment feedback.

### 9 Limitations

#### 9.1 Other LLM Choices

We focused on evaluations based on PaLM-2. Recent advances in LLM agents (e.g., Wei et al., 2022; Yao et al., 2023; Shinn et al., 2023) have shown that different LLMs (e.g., PaLM, GPT-3/4, Codex) generally exhibit a common capacity to benefit from intermediate thoughts and self-criticism. We believe there is a reasonable adaptation of our findings on other LLMs.

#### 9.2 Other Modalities of Input

Large multimodal models can take additional inputs such as screen images, and prior works (e.g., CC-Net (Humphreys et al., 2022)) have shown that extra modality can indeed be beneficial. However,

even with recent designs of large multimodal models, explicit reasoning still takes place in the form of language. Therefore, our proposal could benefit in such multimodal use cases.

### 9.3 Integration Zero-shot Chain-of-Thought

Prior zero-shot works (e.g., Huang et al., 2022; Wang et al., 2023b; Crispino et al., 2023) discovered LLMs can be used to expand prompts with prior knowledge and intermediate steps to work in a zero-shot manner. Theoretically, this line of works can also be integrated into our reflective agent to promote completion rate on the first trial. One potential challenge is that computer control tasks, looking at the input texts, are quite different from those in general domain (e.g., sentiment classification, numerical reasoning). Thus, the quality of extracted prior knowledge needs to be evaluated. We leave this direction to be explore in future work.

### 9.4 Constraining Space for Non-click Actions

In Sec. 5.2, we let the reflection module to interact with the environment, explicitly disabling failed click actions by removing the "id" field on respective elements. This often helps our agent avoid repeating the same mistakes, but only for click actions.

### 9.5 More End-to-end Tasks

Recent few-shot works have used techniques to extract informative reference traces, either from expert or agent exploration (Zheng et al., 2023), to progress more end-to-end computer tasks, such as *book-flight*. We observe such end-to-end tasks remains a significant challenge to *zero-shot* agent.

### 9.6 Higher-order Action Cycle

In Sec. 5, we proposed a structured thought management to facilitate agent's self-reflection. While this module can effectively help LLM agent avoid repeating prior mistakes, there are corner cases need to be covered. In rare cases, we observed the agent can loop over two failed and different traces by accidentally clearing up prior reflection memory. This is because our agent considers reflections on later time steps outdated once there is a reflection entry for earlier time step. Future work can use additional trace memory to avoid such corner cases.

## Acknowledgements

We thank the reviewers of EMNLP for constructive comments and pointers.

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

## A  Prompt for Action Space

Fig. 8 contains the prompt prefix that lines out three action types.

## B  Prompt for Staged Planning

Fig. 9 contains the prompt prefix for maximally plan out actions on each screen in one pass.

## C  Prompt for Action Paraphrase

Fig. 10 contains the prompt prefix for maximally plan out actions on each screen in one pass.

You can generate a series of atomic actions to fulfill a top-level goal. There are three types of atomic actions you can perform. Firstly, you can click an object by referring to its id, such as "click id=...". Secondly, you can enter text to an input field, such as "enter "..." to id=...".' Specifically, you should always wrap the text you want to type in with double quotes. Lastly, you can operate special keys on the keyboard, such as "hold CTRL" and "release CTRL" before and after multiple selections. If dropdown list is available, you can "press ARROWUP x N" or "press ARROWDOWN x N" to press the arrow key N times to iterate over list items, and then "press ENTER" to select the current item.

Figure 8: Prompt that defines three action types.

Now, you need to plan actions that are executable on and only on this screen. For actions that are not executable on this screen, you should leave them to future planning. Your plan should consist of a list of atomic actions on the screen. Please separate them by newline.

Figure 9: Prompt that promotes staged planning.

## D Status Prompt Prefix

Fig. 11 contains prompt component we use to compose reflection prompt for each type of environment feedback.

## E Prompt for Reflection

Fig. 12 contains the prompt prefix for our structured reflection.

## F Completion Rate Tables

We report the completion rate on $43$ tasks in MINI-WOB++ in categories. Performances in *1-screen-1-step* tasks are in Tab. 10, *1-screen-n-step* task in Tab. 11, and *n-screen-n-step* tasks in Tab. 12.

## G Task-wise Statistical Significance

In Tab. 13, we compare $T = 1$ v.s. $T = 5$ on the *n-screen-n-step* tasks. We use the one-tailed McNemar (matched chi-square) for the test.

You are capable of describing actions taken on a computer. The computer screen is represented by the following HTML pseudo code: <screen> {html} </screen> And the action taken is: {action_str} Now, in plain language, please summarize what has been done. You should describe the specific purpose for the action, instead of simply referring to the element id or position of the element. Summary:'

Figure 10: Prompt for paraphrasing an executed action. Newlines are filtered.

STATUS.FAILED: "However, your actions did not complete the goal. Now, you need to identify the earliest critical step where you made a mistake, and suggest a correction."
STATUS.CYCLE: "However, your actions led you to a loop that did not progress the task. Now, you need to identify the earliest critical step where you made a mistake, and suggest a correction."
STATUS.NO_CHANGE: "However, your last action did not cause anything to change on the last screen. You probably used the wrong action type. Now, you need to identify the earliest critical step where you made a mistake, and suggest a correction."
STATUS.IN_COMPLETE: "However, your actions did not finish the task, likely more steps are needed. Now, you need to identify the earliest critical step where you made a mistake, and suggest a correction."
STATUS.IN_PROGRESS: "However, you took too many steps and yet still did not finish the task. Now, you need to identify the earliest critical step where you made a mistake, and suggest a correction."
STATUS.EXCEPTION: "However, your last action is invalid. You should avoid doing that again and try a different action."

Figure 11: Prompt component that describes environment status.

You are operating a computer for a task: {task_name}. You went over a series of screens and executed actions to fulfill a top-level goal. Your action trajectory is as follows: ... The index=k screen: {screen_html at k} Your index=k action: {action at k} ... You conducted the above actions for the top-level goal: goal {status_str} Your suggestion should be in this format: "For action index=A, you should B.", where A is the action index, and B is the suggested action you should have taken. Your suggestion:

Figure 12: Prompt for reflection. Newlines are filtered. *status_str* is defined in Fig. 11.

| Task | T=1 | T=3 | T=5 |
|---|---|---|---|
| click-dialog | 100 | 100 | 100 |
| click-dialog-2 | 100 | 100 | 100 |
| click-link | 100 | 100 | 100 |
| click-button | 100 | 100 | 100 |
| click-tab | 100 | 100 | 100 |
| click-test | 100 | 100 | 100 |
| click-test-2 | 100 | 100 | 100 |
| click-widget | 64 | 96 | 100 |
| focus-text | 100 | 100 | 100 |
| focus-text-2 | 100 | 100 | 100 |

Table 10: Completion rate on 1-screen-1-step tasks.

| Task | T=1 | T=3 | T=5 |
|---|---|---|---|
| click-button-sequence | 100 | 100 | 100 |
| click-checkboxes | 100 | 100 | 100 |
| click-checkboxes-large | 92 | 96 | 100 |
| click-checkboxes-soft | 68 | 84 | 84 |
| click-checkboxes-transfer | 100 | 100 | 100 |
| click-collapsible | 100 | 100 | 100 |
| click-option | 100 | 100 | 100 |
| click-scroll-list | 100 | 100 | 100 |
| enter-password | 100 | 100 | 100 |
| enter-text | 100 | 100 | 100 |
| enter-text-2 | 100 | 100 | 100 |
| enter-text-dynamic | 100 | 100 | 100 |
| find-word | 56 | 72 | 72 |
| login-user | 100 | 100 | 100 |
| multi-layouts | 100 | 100 | 100 |
| multi-ordering | 100 | 100 | 100 |
| read-table | 100 | 100 | 100 |
| read-table-2 | 100 | 100 | 100 |
| social-media-all | 68 | 80 | 80 |
| social-media-some | 96 | 96 | 96 |

Table 11: Completion rate on 1-screen-n-step tasks.

| Task | T=1 | T=3 | T=5 |
|---|---|---|---|
| click-collapsible-2 | 88 | 88 | 88 |
| click-menu-2 | 36 | 72 | 76 |
| click-pie | 96 | 96 | 96 |
| click-tab-2 | 44 | 60 | 68 |
| click-tab-2-hard | 32 | 56 | 64 |
| email-inbox | 96 | 100 | 100 |
| email-inbox-nl-turk | 76 | 92 | 92 |
| email-inbox-forward-nl | 100 | 100 | 100 |
| email-inbox-forward-nl-turn | 100 | 100 | 100 |
| navigate-tree | 96 | 96 | 96 |
| search-engine | 20 | 64 | 72 |
| social-media | 84 | 84 | 84 |
| use-autocomplete | 72 | 96 | 96 |

Table 12: Completion rate on n-screen-n-step tasks.

| Task | #Completion (T=1) | #Completion (T=5) | p-value |
|---|---|---|---|
| click-collapsible-2 | 22 | 22 | - |
| click-menu-2 | 9 | 19 | 0.008 |
| click-pie | 24 | 24 | - |
| click-tab-2 | 11 | 17 | 0.007 |
| click-tab-2-hard | 8 | 16 | 0.0024 |
| email-inbox | 24 | 25 | 0.159 |
| email-inbox-nl-turk | 19 | 23 | 0.042 |
| email-inbox-forward-nl | 25 | 25 | - |
| email-inbox-forward-nl-turn | 25 | 25 | - |
| navigate-tree | 24 | 24 | - |
| search-engine | 5 | 18 | 0.000016 |
| social-media | 21 | 21 | - |
| use-autocomplete | 18 | 24 | 0.007 |

Table 13: Significance test using one-tailed McNemar for each *n-screen-n-step* task.