# OpenReview forum: "A Zero-Shot Language Agent for Computer Control with Structured Reflection"
_EMNLP/2023/Conference — EMNLP 2023 Findings_

### Official Review · Reviewer_HJRY · 2023-07-22

**Soundness:** 3

**Excitement:**

3: Ambivalent: It has merits (e.g., it reports state-of-the-art results, the idea is nice), but there are key weaknesses (e.g., it describes incremental work), and it can significantly benefit from another round of revision. However, I won't object to accepting it if my co-reviewers champion it.

**Paper Topic And Main Contributions:**

This work is a type of input mechanism design for LLM. Since the input length of LLM is limited, the content of the input should be organized more effectively.

**Questions For The Authors:**

Table 5 does not reflect the situation of iterative planning v.s. staged planning very well, but it does show that trying three times is better than trying once?

**Reasons To Accept:**

Similar work on input mechanism design includes process design and prompt design. The author successfully designed a process with CoT and Reflection, enabling LLM to find a feasible action sequence from trial and error.

**Reasons To Reject:**

For the MINIWOB++ environment, the author removed tasks that require visual signals and clicks to complete. The obvious drawback is the lack of image modality information. As of mid-2023, there are not many commercially available LLM APIs or open-source LLMs that can handle large-scale image input capabilities. Once a more powerful multimodal LLM appears, this type of pure text input work will become meaningless.

Furthermore, since this work does not involve LLM training or reinforcement learning, the design of these mechanisms to manipulate computers is limited by the LLM used. In this paper, the author used PaLM2-L, which can only reflect the ability of PaLM2-L in this task. LLM can try and error in the environment, but cannot continuously learn from trial and error to avoid errors in the first attempt next time. This mechanism designs themselves cannot long-term improve LLM's ability to manipulate computers. This makes the article look like just a report for a prompt engineering attempt.

At the same time, some mechanisms designed in the article are not very elegant. For example, to prevent repeating errors that have already occurred in a task, the author "only remove the id attribute, and still allow the element information to be presented to LLM." This approach has a certain degree of randomness and no rationale, and cannot be used as a broad conclusion.

The sections describing the proposed methods in the article, Section 4 and Section 5, are interrupted by one reference article after another. The authors’ original intention was to compare their own methods with others, but this breaks the coherence of reading. In addition, the section proposing the method should add an overall figure to describe the relationship between these subsections and their help in completing different aspects of the task.

**Reproducibility:**

3: Could reproduce the results with some difficulty. The settings of parameters are underspecified or subjectively determined; the training/evaluation data are not widely available.

**Reviewer Confidence:**

3: Pretty sure, but there's a chance I missed something. Although I have a good feel for this area in general, I did not carefully check the paper's details, e.g., the math, experimental design, or novelty.

**Typos Grammar Style And Presentation Improvements:**

Figure 4 and Figure 5 are not vector graphics and some are blurry.
In Table 6, the author attempted to explain whether having LLM reflect on performance could improve it. At most, after trying five times, although it did not surpass the best existing supervised method, it did not perform significantly better than the Few/N-shot method. Can these subtle differences be attributed to the different abilities of LLM itself, and it is difficult to say how much the Reflection Strategies designed by the author play a role in this.

---

> ### Author Rebuttal · Authors · 2023-08-28
>
> Thank you for your detailed review. We believe most of your concerns are not due to the limitations of our work, and they can be adequately addressed by improving our writing in the revision.
>
> > For the MINIWOB++ environment, the author removed tasks that require visual signals and clicks to complete. The obvious drawback is the lack of image modality information. As of mid-2023, there are not many commercially available LLM APIs or open-source LLMs that can handle large-scale image input capabilities. Once a more powerful multimodal LLM appears, this type of pure text input work will become meaningless.
>
> While we understand multimodal LLM can take additional input such as images, we respectfully disagree that our approach "will become meaningless". Our approach for enabling zero-shot capability for agents via structured self reflection is independent of specific input modalities available. Because this is the first work on the topic, we base our work on a language agent because language-based LLM affords a mature base model. Meanwhile, the lack of compelling performance from recent multimodal LLM is a practical limitation, so at the time of this submission, LLM is still a strong choice. Even with recent multimodal design, explicit reasoning still takes place in the form of language. Thus our proposal could benefit such models that are still on the way. We will add such discussion to the final draft.
>
> > Furthermore, since this work does not involve LLM training or reinforcement learning, the design of these mechanisms to manipulate computers is limited by the LLM used. In this paper, the author used PaLM2-L, which can only reflect the ability of PaLM2-L in this task.
> - Our work is actually in the line of the emerging in-context learning and verbal-formed reinforcement learning, such as CoT (NeurIPS22),  ReACT (ICLR23), and Reflexion (arXiv 23). We believe this line of research has been highly valued by the community.
> - There is a shared common capacity to reflect and use intermediate thoughts in recent LLMs. This pattern suggests that our reflection strategy can benefit other LLMs too. While our experiments are focused on PaLM2, the conceptual design of our agent is independent of specific LLM choice.
>
> > LLM can try and error in the environment, but cannot continuously learn from trial and error to avoid errors in the first attempt next time. This mechanism designs themselves cannot long-term improve LLM's ability to manipulate computers. This makes the article look like just a report for a prompt engineering attempt.
> - Long-term or lifetime learning for LLM is still an open question. We definitely did not argue we can solve it in this paper. Our work can be considered a step towards that goal since we showed that an agent model can already learn and improve itself in a short-term period of task execution.
> - Using reflections towards better performance on the first trial is still an open question in the line of reflection works. We genuinely believe the community is on the right path to solve this problem. Firstly, we need to show agents can learn from their own mistakes in computer control problems. Next, we can explore ways to benefit from such mistakes towards better execution on the first trial.
> - Our core contributions (as in the Intro section) is the first zero-shot agent design with structured reflection module. Prompt engineering is part of our work but is not the core.
>
> > At the same time, some mechanisms designed in the article are not very elegant. For example, to prevent repeating errors that have already occurred in a task, the author "only remove the id attribute, and still allow the element information to be presented to LLM." This approach has a certain degree of randomness and no rationale, and cannot be used as a broad conclusion.
> - Our design follows the intuition behind induction heads [1]. Removing the id attribute from the input essentially discourages the LLM to issue actions on that element, therefore the same errors can be better avoided.
> - Removing the entire element could make the screen representation incomplete thus bringing noise and difficulty for LLM to understand the screen context. Our design choice is therefore only to remove the id fields as it is a **minimal** change to the input.
> - We conduct an ablation on the action constraining component below, showing it's a beneficial add-on. We focus on the challenging n-screen-n-step category and the tasks in Fig 7 except the find-word task since it's not centered on click action.
> | Task (num_trial=3) | no constraining action space | with constraining action space |
> |-----|-----|-----|
> | click-menu-2 | 72%  | 72% |
> | click-tab-2 | 56% | 60% |
> | click-tab-2-hard | 52% | 56% |
> | search-engine | 60% | 64% |
>
> [1]: In-context Learning and Induction Heads. Olsson et. al. 22
>
> > The sections describing the proposed methods in the article, Section 4 and Section 5, are interrupted by one reference article after another. The authors’ original intention was to compare their own methods with others, but this breaks the coherence of reading. In addition, the section proposing the method should add an overall figure to describe the relationship between these subsections and their help in completing different aspects of the task.
>
> - Thank you for the suggestions for improving the presentation. There is a wide spectrum of related works for the planning strategy, which we feel important to overview and thus motivate our proposed planning strategy. We will incorporate your suggestions for our revision.
>
> > Question: Table 5 does not reflect the situation of iterative planning v.s. staged planning very well, but it does show that trying three times is better than trying once?
> - Yes num_trial=3 is better than num_trial=1.
> -  The major motivation behind our staged planner is the efficiency (instead of accuracy since both planning strategies ceil at close to 100%). Here, we show our staged planner has substantially better efficiency:
> | | 1st successful trial (25 ex per task) || Up to 3 trials (25 ex per task) ||
> |-----|-----|-----|-----|-----|
> | Task | #actions | Planing call reduction | #action | Planning call reduction |
> | click-checkboxes-large | 234 | 89.7% | 270 | 88.5% |
> | click-checkboxes-soft | 81 | 76.5% | 167 | 61.7% |
> | multi-layout | 175 | 85.7% | 175 | 85.7% |
> | social-media-all | 114 | 82.5% | 224 | 71.4%|
> - The planning call reduction is calculated by comparing the number of staged planner calls vs the number of iterative planner calls. The number of iterative planner calls is essentially the number of actions since each action requires a round of LLM inference. Clearly, our staged planner offers a substantial planner call reduction. Thus, it is efficient as we claimed. We will add this result to the final draft to better motivate it.
>
> >  In Table 6, the author attempted to explain whether having LLM reflect on performance could improve it. At most, after trying five times, although it did not surpass the best existing supervised method, it did not perform significantly better than the Few/N-shot method. Can these subtle differences be attributed to the different abilities of LLM itself, and it is difficult to say how much the Reflection Strategies designed by the author play a role in this.
> - Our proposed reflection strategy plays a substantial role. This comparison is essentially the num_trial>1 vs num_trial=1 across Sec 6. We also adapted the original verbal reflection to compare with our structured thought management. Results in Fig 7 suggests that our structured reflection strategy is a beneficial add-on.
> - We also want to point out that moving from few/N-shot to zero-shot is a substantial challenge. We presented performances from supervised and few/n-shot agents to show where our agent is on the benchmark, however, they are not in direct comparison since the strategies are different.

---

### Official Review · Reviewer_7sGp · 2023-08-03

**Soundness:** 4

**Excitement:**

4: Strong: This paper deepens the understanding of some phenomenon or lowers the barriers to an existing research direction.

**Paper Topic And Main Contributions:**

This work investigates the use of a PLM (PaLM-2) for computer control tasks (MiniWOB++) in a zero-shot setting (no further training).

The authors find in a manual evaluation of 13 MiniWOB++ tasks/environments that at least 5 of these could be solved by using information that is contained in the HTML (of the webpage) although the information is not visible (in the browser). Thus the authors suggest to expand elements in the HTML only when they are actually accessible in the UI (still text is the only modality) to make the tasks more aligned with human perception (and more difficult for LLMs).

In addition the authors
- (a) suggest to produce actions only for the screen at hand ("staged plan-and-follower", in contrast to other work that produces actions for the whole problem) and
- (b) propose a programmatic extension to the Reflexion approach (Shinn et al. 2023) so that actions (click element with id, insert text to element with id, press key) that have lead to unsuccessful outcomes are disabled for later trials.

They test their modifications with the PaLM-2 model for different number of trials (1, 3, 5) on a sub-selection of 43 MiniWOB++ environments (excluded are those that require vision) and compare the success rates with other works (WebN-T5, CC-Net, RCI, AdaPlanner). The authors find that their approach can solve easy (1-screen:1-step) and medium (1-screen:n-step) environments in all cases (except an ambiguous case). And is performing similarily to few-show approaches (RCI, AdaPln) for the hard (n-screen:n-steps) environments. Still, the supervised approach (CC-Net) is reaching the best performance.

**Questions For The Authors:**

- Question A: You describe the 3 action types, but how many actions are on average necessary to solve an environment? Does 3 trials performance improve (in some cases) so drastically, because maybe only a (correct) single action is left after 2 trials?
- Question B: Does the good performance of the programmatic reflexion approach mean that this is what current models lack? A memory module that keeps track of what is going wrong during an interaction?

**Reasons To Accept:**

- a well-written and easy to follow study on zero-shot usage of a pre-trained LLM for computer control tasks
- manual evaluation of the existing MiniWOB++ tasks which reveal problems in the way these environments have been approached before
- the paper fits nicely into the (well explained) research field of computer control and indicates further directions for the community ("thought management")

**Reasons To Reject:**

- the contribution would be stronger, when the "thought management" (excluding failed actions) would be learnt (and not programmed)
- only results for a single proprietary model (PaLM-2) and none for open-access models (from e.g. huggingfaces)

**Reproducibility:**

3: Could reproduce the results with some difficulty. The settings of parameters are underspecified or subjectively determined; the training/evaluation data are not widely available.

**Reviewer Confidence:**

3: Pretty sure, but there's a chance I missed something. Although I have a good feel for this area in general, I did not carefully check the paper's details, e.g., the math, experimental design, or novelty.

**Typos Grammar Style And Presentation Improvements:**

- L196: actions could be defined more precisely (or pointed to the appendix)
- L445: "the 9 tasks" -- better "9 tasks" because this confuses with Table 4 where 10 tasks are indicated
- L490: "out" -- "our"
- 7.3. social media task could need a better task description (what has to be done) to better understand the numbers in Table 7
- Table 7 is not intuitively to read (10+?), captions more elaborate on this
- performance should be somehow weighted by the number of necessary trials (or are always alls trials used?)
- The proposed approach reminds mind me of Hierarchical RL: Learning a (sub-) policy for each room (screen) in an environment

References:
- could need another round for camera-ready (resolve arxiv refs if possible)

Reproducibility:
- seeds are not reported
- I assume Palm-2 was accessed via an API, so the version and/or date should be reported

---

> ### Author Rebuttal · Authors · 2023-08-28
>
> Thank you for your encouraging remarks and insightful feedback. We believe addressing the points you raised will improve the clarity our paper and strengthen our contributions in the revision.
>
> > the contribution would be stronger, when the "thought management" (excluding failed actions) would be learnt (and not programmed)
> - We do agree that a more end-to-end neural model for thought management is very interesting. Our work essentially shows that proper thought management is a beneficial add-on component. Thus, learning such neural module can be the next step. Such neural module can inform LLM towards finishing a given task in one trial. This direction still an open question in the line of reflection works.
>
> > Question A: You describe the 3 action types, but how many actions are on average necessary to solve an environment?
> - We coarsely bucket the expected number of actions to finish a task into [0, 4), [4, 7), and [7, +). In the below table, we show that there is indeed a pattern between action length and task completion rate.
> (Note that 1-screen-1-step tasks are omitted since they only require 1 action.)
> | | <=4 | [4, 7) | >=7 |
> |-----|-----|-----|-----|
> | task | 13 | 15 | 5 |
> | Completion rate | 94% | 91.6% | 90.4% |
> - However, we should also note that required action length is only one of the factors in task complexity. For instance, in the search-engine task, an average of 6 steps is required. But the agent performance is substantially lower than the click-checkboxes-large task which requires 10+ steps to finish. This is due to search-engine involving unseen screens so that agent has to explore and plan on the fly.
>
> > Questions A: Does 3 trials performance improve (in some cases) so drastically, because maybe only a (correct) single action is left after 2 trials?
> - We believe this is not the case since that, during each trial, our agent will maximally attempt on the task until a failure signal is raised. It does not stop in the middle of execution if actions are properly grounded on the environment.
> - The drastic improvements come from our reflection module. The agent learns (verbally) to identify prior errors and correct them across multiple trials.
>
> > Question B: Does the good performance of the programmatic reflexion approach mean that this is what current models lack? A memory module that keeps track of what is going wrong during an interaction?
> - We genuinely think so too. A helper memory management module for LLM is essentially what we proposed in the draft.
> - A future direction is to benefit from prior errors and corrections for better completion rate in one trial. And yes, this can be potentially managed by a separately trained memory module. In general, it is an interesting and still open question in the line of reflection works.
>
> > References: could need another round for camera-ready (resolve arxiv refs if possible)
> - We will improve citations in the revision.
>
> > Reproducibility: seeds are not reported
> - As mentioned in Sec 6, we use seeds 1000-1024 following prior works (e.g., Pix2Act) to start seeds from 1000 for testing.
>
> > Reproducibility: I assume Palm-2 was accessed via an API, so the version and/or date should be reported
> - We will report the version of PaLM and date it was ready for access. We want to emphasize that, while different LLMs are known to behave differently, our proposed method is conceptually orthogonal to LLM choices.
>
> > Are always all trials used?:
> - Not aways. The number of trial is the maximum trials our agent will attempt on a task. Once a task is completed by a trial, later trials will not be conducted.

---

### Official Review · Reviewer_wto2 · 2023-08-07

**Typos Grammar Style And Presentation Improvements:** N/A
**Soundness:** 4

**Excitement:**

3: Ambivalent: It has merits (e.g., it reports state-of-the-art results, the idea is nice), but there are key weaknesses (e.g., it describes incremental work), and it can significantly benefit from another round of revision. However, I won't object to accepting it if my co-reviewers champion it.

**Missing References:**

N/A

**Paper Topic And Main Contributions:**

This paper presents a systems contribution in a novel LLM-based agent for interacting with computer environments (specifically, internet web pages a la the MiniWoB++ benchmark) zero-shot, solely learning through the feedback received from the web interface/sparse task success.

The proposed approach built on top of the PaLM-2 language model is novel in that it requires no finetuning or in-context examples to perform tasks, and is able to obtain high task success rate through a combination of the following contributions:

- A compact (reparameterized) screen representation that simplifies the HTML visible by the LLM as context.
- A staged action planner that allows the LLM to output *multiple* actions at a given timestep/context in a way that enables better temporal consistency.
- A structured reflection module that uses an external structure to logically prune out parts of historical action traces whenever a task is failed, aiding in replanning the next task attempt.

While a generally complex paper to follow (echoing the complexity of the underlying system), the results show that the proposed approach can not only obtain high success rates on “simple” tasks from MiniWOB++ (requiring only a single step to solve), but that the structured reflection + staged planning components enable strong performance on more complex tasks (more contextual, more turns), beating or coming close to the performance of baseline approaches that are trained with supervision or are enriched with few-shot examples.

**Questions For The Authors:**

A. Algorithm 1 is very very hard to follow; can we add a more annotated version of the algorithm (or better, a visual that captures exactly what’s happening in the structured reflection module)?

B. It’s not immediately intuitive why the staged action prediction is better than closed loop (predict action iteratively, one after the other) in this approach? Can you provide more intuition or motivating examples that show when closed-loop action prediction fails, and staged action prediction succeeds?

**Reasons To Accept:**

The systems contribution in this work is strong; each of the proposed components is well-motivated (if not properly ablated), and the core nugget in the results — that the proposed agent can solve these tasks without finetuning or in-context examples — is clearly impressive and a step forward.

I think this work also does a great job of codifying the prior work in this area, and different assumptions made in building LLM-based web agents.

**Reasons To Reject:**

As hinted at above, I think there are several design decisions that are not properly ablated. One general criticism I have for recent work in LLM-based agents that rely on prompting approaches is verifying that new methods work for the same LLMs used in prior work (where possible). This work makes the choice to use PaLM-2; however, all prior methods cited as baselines in the tables are (I believe?) not run with PaLM-2, instead reporting results from the original papers, and original LLMs (e.g., GPT-3.5, text-davinci-002). Indeed, given PaLM-2’s general lack of availability to the larger community, it’s not clear at all whether the gains come from the underlying LLM’s capability, or from the contributions described above.

Following the above general criticism, it would be nice to ablate specific components of the proposed approach vs. existing work. How would a method like RCI work if it used the structured reflection module instead of the naive single-step reflection feedback? How would AdaPlanner perform with the staged action prediction component?

It feels as if this is a paper that is presenting a series of “modules” that can be grafted on to many existing LLM-based web agents; if the contribution of this paper (as I see it) are these modules, then more work needs to be done to verify that these modules are generally helpful in settings beyond just the current system implementation.

---

As a minor weakness — I’m not sure that I agree about the “compact” screen representation relaxing assumptions on the environment, or forcing the agent to behave more/less rationally when given limited information. I think these claims need to be tempered a bit in the paper — the existing LLM agents are already “unnatural” in that they are operating over HTML vs. visual screens... not sure we should further restrict what HTML is “valid/invalid” for an agent to see (dumping page source, depending on webpage, can result in many different plausible elements). I think it's better instead to scope down this claim to the results observed in evaluating the proposed system with it’s compact representation.

---
EDIT (Post-Rebuttal): Given the authors clarifications and extra discussion provided during the rebuttal, I am revising my soundness score upwards to a 4!

**Reproducibility:**

2: Would be hard pressed to reproduce the results. The contribution depends on data that are simply not available outside the author's institution or consortium; not enough details are provided.

**Reviewer Confidence:**

4: Quite sure. I tried to check the important points carefully. It's unlikely, though conceivable, that I missed something that should affect my ratings.

---

> ### Author Rebuttal · Authors · 2023-08-28
>
> Thank you for your detailed comments and constructive feedback! We address your comments here and will revise our paper accordingly.
>
> > As hinted at above, I think there are several design decisions that are not properly ablated.
> > ...
> > Following the above general criticism, it would be nice to ablate specific components of the proposed approach vs. existing work. How would a method like RCI work if it used the structured reflection module instead of the naive single-step reflection feedback? How would AdaPlanner perform with the staged action prediction component?
> - It is nontrivial to conduct strictly clean comparison against prior works, and we seriously considered a number of options for doing so. One observation of recent LLM-based works on the problem (including our own) is that each component is tailored to work within the proposed paradigm. This includes RCI, Adaplanner, ours, and other cited works; let alone to say that our work is the first zero-shot agent which is quite different from prior few-shot ones. Therefore, our ablation studies were conducted within our proposed paradigm.
> - To illustrate, let us consider the component differences between RCI and ours in details:
> | | Number of shots | Consistent screen | Planner | #Reflection |
> |-----|-----|-----|-----|-----|
> | RCI | Few to many | No | Iterative | 1 |
> | Ours | 0 | Yes | Staged | N |
>   - **On the number of shots and screen consistency**. There is likely no easy approach to reduce RCI from few-shot to zero-shot. This correlates to their use of inconsistent screen representations. We manually inspected RCI’s shots on their Github repo and found that the agent relies on those given demonstrations to predict the id of an element on an unseen screen, so that the generated action can be properly grounded on the environment. That is, the existing use of few-shot prompts relies on the use of inconsistent screens. These two components are bonded together.
>   - **On the choice of planner**. There might be performance differences to tell but the completion rate on 1-screen-n-step tasks is already saturated, the gap is likely minimal. Our argument on this component is actually to simplify the process: staged planning is already good enough when you need multiple steps on a single screen, so one doesn't need iterative planning due to the efficiency cost (evidence is in our response to your question B).
>   - **On the rounds of reflection**. Broadly speaking, RCI could align to our agent with at most 1 round of reflection while our agent can do more rounds. Beyond this difference is the structured reflection (ours) vs plain reflection (RCI). On this component, we already have ablation study in the paper and the performance gap is substantial (i.e., num_trial=1 vs num_trial=3 or 5).
>   - **On the reflection strategy**. We have presented ablation study against an adaptation of the original reflection (from Reflexion 2023 paper) in Fig 7. The conclusion is that our structured thought management is a beneficial add-on component.
>
> > Indeed, given PaLM-2’s general lack of availability to the larger community, it’s not clear at all whether the gains come from the underlying LLM’s capability, or from the contributions described above.
> - Recent advances in LLM agents (e.g., CoT (NeurIPS22), ReACT (ICLR23), and Reflexion (arXiv 23)) have shown that different LLMs (i.e., PaLM, GPT-3/4, Codex) generally exhibit a common capacity to benefit from intermediate thoughts and self-reflection. We believe there is a reasonable adaptation of our findings on other LLMs.
> - There are probably some extents of prompt dependency on LLM. We believe this is expected since switching LLMs sometimes requires adapting prompt/instruction. But we want to emphasize that the proposed methodology is conceptually orthogonal to LLM choices.
>
> > It feels as if this is a paper that is presenting a series of “modules” that can be grafted on to many existing LLM-based web agents; if the contribution of this paper (as I see it) are these modules, then more work needs to be done to verify that these modules are generally helpful in settings beyond just the current system implementation.
> - We believe the answer is it depends on the use case. This work is actually **the first zero-shot agent** in the line. Our design considerations are focused on zero-shot use case which is different from prior few-shot works.
> - With recent LLMs generally showing a degree of capacity to reflect and use intermediate thoughts, we believe our structured reflection has a good promise to generalize across different LLMs.
>
> > As a minor weakness — I’m not sure that I agree about the “compact” screen representation relaxing assumptions on the environment, or forcing the agent to behave more/less rationally when given limited information.
> - We will improve the presentation. The motivation mainly has 3 aspects:
>   1) The compact representation is essentially a simplification of the original HTML which tends to be long and verbose. This poses a practical limitation to existing LLM input length. In our case, reflection requires seeing the complete trajectory that could span over multiple screens. This often runs out the input length limit for PaLM-2. Thus, shorter screen code is preferred.
>   2) Following [1], compact screen HTML codes can give compelling results on UI understanding tasks, suggesting it’s a good alternative. We have this citation in the reference.
>   3) Lining up the separation of valid/invalid screen HTML is important for evaluation. As we have seen that many of the MiniWoB++ tasks are already easy for today’s LLM. Exposing more unseen elements risks hiding the actual challenge in the navigation tasks. For instance, exposing unseen elements basically simplifies n-screen-n-step tasks into 1-screen-n-step ones. However, our experiment shows that n-screen-n-step ones are actually much harder to deal with.
>
> [1]: Enabling Conversational Interaction with Mobile UI using Large Language Models. Wang et. al. CHI 2023
>
> > Question A. Algorithm 1 is very very hard to follow
> - We will add an annotated version of Algo 1 to the draft. The algorithm works in an intuitive way. It starts with a loop over a number of trials. At each trial,
>   - Line 4: it checks if there is a reflection statement from previous trials.
>   - Line 5-6: If there is any, it identifies if there is a suggested action to heuristically take, or sample it from LLM.
>   - Line 7: Such action is executed on the environment. If an error occurs,
>   - Line 8: it starts to reflect by observing the entire action trajectory, and output a statement consists of (a, a’).
>   - Line 9-10: the wrong action is recorded and heuristically disabled on the environment, if applicable.
>   - Line 11: the verbal form of reflection statement is also recorded, so that LLM can use it as input during planning.
>   - Line 12: with a new action a’ suggested for timestep j, previous suggestions after timestep j can be considered outdated, thus memory is cleared accordingly.
>
> > Question B. It’s not immediately intuitive why the staged action prediction is better than closed loop (predict action iteratively, one after the other) in this approach? Can you provide more intuition or motivating examples that show when closed-loop action prediction fails, and staged action prediction succeeds?
> - For intuition on concrete examples, we recommend to try out this [demo](https://miniwob.farama.org/environments/click-checkboxes-soft/). We want to clarify that the major motivation of our staged planning is actually **efficiency** instead of accuracy.
>   - In iteratively planning, an agent will need to query LLM for each click on the screen.
>   - In our staged planning, all items to select are issued in one query.
>   - These two approaches both ceil at ~100% on 1-screen-n-step tasks. Therefore we propose to do planning more efficiently.
> -  Here, we show our staged planner has substantially better efficiency:
> | | 1st successful trial (25 ex per task) || Up to 3 trials (25 ex per task) ||
> |-----|-----|-----|-----|-----|
> | Task | #actions | Planing call reduction | #action | Planning call reduction |
> | click-checkboxes-large | 234 | 89.7% | 270 | 88.5% |
> | click-checkboxes-soft | 81 | 76.5% | 167 | 61.7% |
> | multi-layout | 175 | 85.7% | 175 | 85.7% |
> | social-media-all | 114 | 82.5% | 224 | 71.4%|
> - The planning call reduction is calculated by comparing the number of staged planner calls vs the number of iterative planner calls. The number of iterative planner calls is essentially the number of actions since each action requires a round of LLM inference. Clearly, our staged planner offers a substantial planner call reduction. Thus, it is efficient as we claimed. We will add this result to the final draft to better motivate it.

---

### Official Review · Reviewer_zQff · 2023-08-12

**Soundness:** 2

**Excitement:**

2: Mediocre: This paper makes marginal contributions (vs non-contemporaneous work), so I would rather not see it in the conference.

**Paper Topic And Main Contributions:**

This paper introduces a zero-shot LLM agent designed for web interaction tasks. Enhanced with a self-reflection and structured thought management module, the agent can generate actionable plans in a staged manner. Experimental results on the MINIWOB++ benchmark reveal that while the proposed method matches state-of-the-art (SOTA) performance in simpler tasks, it tends to fall short in more complex ones.

**Reasons To Accept:**

The exploration of differences between interactive and staged planning is compelling.

The study presents a novel approach to enhancing LLM agent planning in real-world interactive tasks.

**Reasons To Reject:**

The rationale behind using a staged planner with LLMs for the give task could be better articulated. For instance, could RL agents or classic planners tackle MINIWOB++ with interactive planning?

Experimental controls for random noise appear insufficient. Reporting standard deviations, repetitions of measurements, or carrying out statistical analyses would strengthen the work.

On line 465, the authors posit that the staged planner requires fewer LLM queries than the interactive planner, yet fail to provide supporting experimental evidence.

Both the Structured Thought Management and Constraining Action Space seem to boost performance alongside the number of trials. Ablation studies to individually assess their impacts would be beneficial.

In the results for 1-screen-1-step and 1-screen-n-step tasks, there's a noticeable ceiling effect, with most tasks achieving a 100% completion rate. This could bias average performance comparisons across models. Focusing analyses on n-screen-n-step tasks' results might offer a more balanced perspective.

**Reproducibility:**

4: Could mostly reproduce the results, but there may be some variation because of sample variance or minor variations in their interpretation of the protocol or method.

**Reviewer Confidence:**

2: Willing to defend my evaluation, but it is fairly likely that I missed some details, didn't understand some central points, or can't be sure about the novelty of the work.

**Typos Grammar Style And Presentation Improvements:**

The blending of the proposed methodology with prior research is confusing. I would suggest putting previous work in a related work section, and describing your own work in a method section.

The task environment should be delineated more than just referencing the MINIWOB++ paper. Terms like "screens," "candidates," "trials," and "steps" are used in sections 4 and 5 without prior introduction.

The adapted comparison mentioned in line 508 is unclear. Could the authors provide further details or clarification?

---

> ### Author Rebuttal · Authors · 2023-08-28
>
> Thank you very much for your thorough reviews. We address your concerns here, and we believe all of your questions can be addressed in the revision.
>
> > Experimental results on the MINIWOB++ benchmark reveal that while the proposed method matches state-of-the-art (SOTA) performance in simpler tasks, it tends to fall short in more complex ones.
> - Our work presents the first *zero-shot* agent for computer control problems. Zero-shot agents inherently face challenges due to the lack of expert demonstrations, thereby generally expected to underperform supervised or few-shot approaches. Nonetheless, our method impressively achieved state-of-the-art results across a variety of tasks, highlighting the significance of our contribution. Our future work will further explore zero-shot agents for more complex tasks.
>
> > The rationale behind using a staged planner with LLMs for the give task could be better articulated.
> - We will improve the presentation. The main motivation of using a staged planner is efficiency. In our experiments, staged planner performs on par with prior iterative planner approaches while our query efficiency is substantially reduced (evidences are in the “Efficiency” bullet point).
>
> > Experimental controls for random noise appear insufficient. Reporting standard deviations, repetitions of measurements, or carrying out statistical analyses would strengthen the work.
> - We use 0 temperature during decoding, following prior works on MiniWoB (e.g., RCI) to have deterministic and reproducible results.
> - Agent performances are reported as the average of 25 randomly instantiated examples for each task. Even for the same task, different random instantiations could have different complexities, and each instantiation is evaluated in either 1 (for successful completion) or 0 (incomplete for any reason). Thus measuring standard deviation of completion rate does not tell much.
> - For statistical significance over the number of trials, we ran the t-test implemented in [1] over 13 n-screen-n-step tasks as a whole (13x25=325 examples in total). The results are indeed significant (p<0.05):
>   - From num_trial=1 to 3, p-value is 2e-10
>   - From num_trial=3 to 5, p-value is 0.002
>   - From num_trial=1 to 5, p-value is 2e-12
>   - We will add these analyses to the draft.
>
> [1]: The Hitchhiker’s Guide to Testing Statistical Significance in Natural Language Processing. Dror et. al. (ACL18)
>
> > On line 465, the authors posit that the staged planner requires fewer LLM queries than the interactive planner, yet fail to provide supporting experimental evidence.
> - We use the 1-screen-n-step category to demonstrate the efficiency improvement. We select tasks that require relatively longer action traces (>= 7 actions) on a single screen. Specifically, we experiment with num_trial=3 and count the planning call reduction for successfully completed traces as well as including failed traces. The result is measured for 25 examples for each task:
> | | 1st successful trial (25 ex per task) || Up to 3 trials (25 ex per task) ||
> |-----|-----|-----|-----|-----|
> | Task | #actions | Planing call reduction | #action | Planning call reduction |
> | click-checkboxes-large | 234 | 89.7% | 270 | 88.5% |
> | click-checkboxes-soft | 81 | 76.5% | 167 | 61.7% |
> | multi-layout | 175 | 85.7% | 175 | 85.7% |
> | social-media-all | 114 | 82.5% | 224 | 71.4%|
> - Specifically, the planning call reduction is calculated by comparing the number of staged planner calls vs the number of iterative planner calls. The number of iterative planner calls is essentially the number of actions since each action requires a round of LLM inference. Clearly, our staged planner offers a substantial planner call reduction. **Thus, it is efficient as we claimed.** We will add this result to the final draft.
> (The numbers are the same for task multi-layout, since our agent achieved 100% completion rate on the first trial.)
> - We want to clarify that the argument we want to make on staged planning is that, with such large efficiency improvement and similar completion rate (both staged and iterative ones ceiling at ~100%), why not just use the more efficient one.
>
> > Both the Structured Thought Management and Constraining Action Space seem to boost performance alongside the number of trials. Ablation studies to individually assess their impacts would be beneficial.
> - Here we do an A/B test on the action constraining component. Specifically, we ablate whether to deterministically remove the element id from HTML so that the agent will avoid issuing actions on it. The below result shows that heuristically constraining action space (in addition to pure verbal instruction) indeed helps the agent model.
> - We use challenging tasks in Table 7, except the find-word task since it’s not centered on click action. Completion rate is measured over 25 examples for each task.  Note that action constraining is a small module in our agent construction thus the relative small improvement is expected. We will add this result to the draft.
> | Task (num_trial=3) | no constraining action space | with constraining action space |
> |-----|-----|-----|
> | click-menu-2 | 72%  | 72% |
> | click-tab-2 | 56% | 60% |
> | click-tab-2-hard | 52% | 56% |
> | search-engine | 60% | 64% |
> - For ablation study on the structured thought management, we already have a comparison in Fig 7 showing that the thought management is a beneficial add-on.
>
> > In the results for 1-screen-1-step and 1-screen-n-step tasks, there's a noticeable ceiling effect, with most tasks achieving a 100% completion rate. This could bias average performance comparisons across models. Focusing analyses on n-screen-n-step tasks' results might offer a more balanced perspective.
> - This is why we separate the testing on different complexity category, unlike prior works that average over all tasks which included a lot of easy tasks.
> - Reporting results on easy tasks is essential to support our claim that a staged planner is enough for easy tasks. Prior works on these types of tasks exhibit similar ceiling effects, therefore matching similar performance is the bottom line.
> - Our proposal of the structured reflection is indeed focusing on more challenging tasks (i.e., n-screen-n-step).
>
> > The adapted comparison mentioned in line 508 is unclear. Could the authors provide further details or clarification?
> - We will improve the presentation on this ablation study. The two variants for comparison are:
> - **Structurally managed timestep**: The original reflection mechanism (as introduced in the Reflexion paper) inserts reflection memory entries to the planning prompt regardless of what timestep the identified error occurred. In our case, we only insert the memory at the corresponding time step so that actions before this time step can be deterministically repeated for better efficiency.
> - **Structurally managed thoughts**: On top of the structurally managed timestep, we also manage the expiration of thoughts over multiple trials, as well as constraining the action space associated with the thoughts. This is essentially Algo 1.

---

### Meta-Review · Area_Chair_w4Mu · 2023-09-18

**Recommendation:** 3

**Metareview:**

Strengths of the paper include the systems contribution with well-motivated individual components. The paper is also written well and clear and the reviewers thought the ideas themselves are useful in guiding the field and discovering novel problems. A potential drawback of the paper is that it was conducted with a proprietary LLM rather than an open-source one which would make it difficult to reproduce results.

---

### Decision · Program_Chairs · 2023-10-07

**Decision:**

Accept-Findings

**Comment:**

Strengths of the paper include the systems contribution with well-motivated individual components. The paper is also written well and clear and the reviewers thought the ideas themselves are useful in guiding the field and discovering novel problems. A potential drawback of the paper is that it was conducted with a proprietary LLM rather than an open-source one which would make it difficult to reproduce results.